# Identifying Barriers to Basic Needs, Academic Success, and the Vaccination Pattern among College Students during the COVID-19 Pandemic

**DOI:** 10.3390/ijerph20206924

**Published:** 2023-10-14

**Authors:** Subi Gandhi, Ryan Glaman, Alexandra Jordan, Dylan DiChristofaro, Katelyn Clark, Viraj Gandhi, Jade Smith

**Affiliations:** 1Department of Medical Lab Sciences, Public Health, and Nutrition Science, Tarleton State University, 1333 West Washington, Stephenville, TX 76402, USA; katelyn.clark@go.tarleton.edu (K.C.); jade.smith@go.tarleton.edu (J.S.); 2Department of Educational Leadership & Technology, E.J. Howell Building 404, College of Education, Tarleton State University, 1333 West Washington, Stephenville, TX 76402, USA; glaman@tarleton.edu; 3Department of Epidemiology, Human Genetics, & Environmental Science, School of Public Health, The University of Texas Health Science Center, 1200 Pressler Street, Houston, TX 77030, USA; alexandraleighjordan@gmail.com; 4Department of Counseling, Tarleton State University, 10850 Texan Rider Dr., Fort Worth, TX 76036, USA; dylan.dicristofaro@go.tarleton.edu; 5Robbins College of Health and Human Services, Baylor University, Hankamer Academic Center, 1428 S 5th Street, Waco, TX 76706, USA; viraj_gandhi1@baylor.edu

**Keywords:** pandemic, college health, colleges, academic barriers, socioeconomic barriers, vaccine attitudes

## Abstract

College students face significant challenges during large-scale disease outbreaks that potentially compromise their basic needs, vaccine confidence, and academic success. Using a cross-sectional design and convenience sampling, we examined the impact of COVID-19 among college students (*N* = 828). The survey was administered using the Qualtrics survey platform to collect data on multiple demographic and health behaviors of students in the summer semester (2021). Our study demonstrated that the most common financial shock experienced by the study participants was job loss, with less remarkable changes in food and housing insecurities. Academically, students had the most difficulty learning online compared to other modalities (face-to-face, Hyflex, etc.) and struggled with staying motivated. They also struggled with group work and finding appropriate learning spaces. However, many did not use university support systems such as career and tutoring services. Exploring the COVID-19 vaccination attitudes, we found that only age, ethnicity, classification, and health insurance status were associated with getting vaccinated (*p* < 0.05). When the learning environment was assessed for various modalities, only college attendance was significantly associated (*p* < 0.05) with the accessible platforms (online, Hyflex, face-to-face, and others); however, nearly 40% of students reported difficulty learning on an online learning platform compared other categories that had much lower proportions. Our findings underscore an immediate need for universities to take measures to improve their preparedness and response strategies to mitigate the negative effects of future large-scale public health emergencies among students.

## 1. Introduction

Following its discovery in late 2019, COVID-19 evolved into a global public emergency that rapidly supplanted the 1918 influenza pandemic as the most lethal respiratory pandemic in history [1]. As of 20 July 2023, SARS-CoV-2 has infected 750 million and killed nearly 7 million people globally, with the United States making up over 100 million cases and 1.1 million deaths, respectively [2,3].

The accelerated spread of illness since the beginning of 2020 prompted the closure of many schools and universities nationwide, creating swift and distressing challenges for students of all ages. In particular, prior to the beginning of the pandemic, students on college campuses were in the midst of the liminal space of young adulthood. This vulnerable period of burgeoning independence is marked by shifting career possibilities, residences, and social dynamics within friend and family groups. When colleges announced the closure of on-campus housing, more and more students were forced to contend with direct threats to their emerging autonomy: food and housing insecurity, unemployment, and an abrupt transition to virtual and hybrid learning platforms [4,5]. These issues reverberated through all students, creating staggeringly disparate impacts. While there has been much focus on the impact of pandemic-era changes on students, the extent of the vulnerabilities experienced by college students is worth investigating, as these experiences can have a lasting impact on their adult lives.

### 1.1. Barriers to Basic Needs

Even beyond the context of a global public health emergency, the correlation between financial insecurity and poor academic outcomes is well documented [6]. Pandemic-era closures and changes have been an exacerbating force, leaving many students without jobs or at least with reduced wages or hours. Over 13% of higher education jobs were cut during the pandemic, including student jobs on campuses [5], and in general, among students with jobs, 38% lost work due to cancellations [7]. This was especially prevalent in lower-income, female, LGBTQ (Lesbian, Gay, Bisexual, Transgender, and Queer/Questioning), and BIPOC (Black, Indigenous, People of Color) students [8,9,10].

Furthermore, many students, although predominantly lower income, Black, and Hispanic students, were also required to risk working in high-exposure environments [6]. Some students became the sole income provider if they had moved back home or assumed caretaker roles in their family, which further reduced their income and reliable access to food [4,7]. BIPOC, LGBTQ, first-generation, and disabled students have been among the most impacted populations to experience financial hardship due to job loss, reduced job opportunities, financial aid loss, family member job loss, and increased living expenses [7].

The protective factors associated with campus living—that is, social safety nets that in part negate home socioeconomic status, like reliable access to healthy food, housing, technology, and employment—became abruptly unavailable following campus closures [7]. As it stood in 2018, nearly half of all college students lived at or below the poverty line [8], and minority students were more likely to be left without access to shelter, food, and safety [7]. These students were at significantly higher risk of dropping classes to preserve their GPA and financial aid [9], as well as dropping out following the sudden loss of access to basic resources needed for academic success, such as reliable internet connection and laptops [7].

### 1.2. Barriers in the Learning Environment

Technological inequity is a far-reaching and complex issue [10], but the transition to online learning created new barriers to academic success [11]. With difficulty affording up-to-date technology, students are more likely to experience hardware and software issues [12], and as many as 37% of students lack access to the internet at home [13]. Many more rely on cellular hotspots that can often only provide suboptimal internet speed and quality [12]. Together, connectivity issues combined with a lack of technology create a cycle that culminates in frustration, shame, lower GPA, and a higher risk of dropping out [12,14].

It is well documented that as in-person learning dwindled, the massive shift to online instruction struck a deep blow to students’ confidence in completing the academic year successfully. Sweeping changes to teaching procedures, considerable increases in workload, unclear instructions about class expectations, and a lower quality of education all significantly damaged students’ motivation and perceived self-efficacy [15]. Many students noted a lack of communication from professors [16], and reported feelings of isolation and abandonment [14]. Compounded with frequent worries about a “world in chaos” [17] and a home environment that may or may not be conducive to learning [12], students exhibited a greater tendency towards procrastination and distraction and were disengaged from the classroom [16].

Paradoxically, some students were more engaged and increased their study time [18], but these effects were predicted by a student’s position in the socioeconomic divide. Low-income students not only experienced greater learning difficulties but also delayed graduation, changed their major, dropped a class, and dropped out of college at significantly higher rates than their middle-income and upper-income peers [18]. Prowse et al. found that one-third of students felt that the shift to virtual learning was difficult, and just under one third reported adverse effects on their academic success [19]. Complicating matters further, ElTohamy et al. reported that students who remained on campus during the 2021 spring semester reported higher levels of distress than those living off campus, though there are mixed findings as to whether or not living off campus was truly protective [20]. In fact, while students living with parents were less likely to struggle with food insecurity, work, stress, and health [20], they were at higher risk of being exposed to abuse, especially if their family did not respect their sexual or gender identity [7]. Following campus closures, one in ten students relocated to environments where they experienced abuse, and one in twenty students relocated to a home in which they did not feel safe and protected [7].

Small, insulated communities made for isolation and disturbed social lives, which, in turn, negatively impacted students’ mental health [14]. Combined with a high risk of long COVID, or even death, many college students contended with shifting family dynamics that accompanied taking on primary income earning or caretaking roles [7]. These distractions and stressors made it increasingly difficult to focus on academic work, compromising their success.

### 1.3. Vaccine Hesitancy

Although COVID-19 vaccination among college students is a crucial component to slowing the infection rate, vaccine hesitancy threatens community-wide protection [21,22]. There are many dimensions to the variation in vaccination rate among this vulnerable population, but distrust, fear of side effects, and misinformation seem to fuel vaccine hesitancy [21].

Across the country, effective non-pharmaceutical interventions, such as masking and social distancing, had varying degrees of participation [23,24]. The vaccine was developed swiftly and efficiently as a means of ending the pandemic, but its development and roll-out became the topic of fierce political debate across America [25]. College students were no exception in this controversy, and as a result, vaccine uptake also became a major challenge in this vulnerable group.

Anti-intellectualism, a generalized suspicion of experts and empirical evidence [26], negatively influenced the risk perception associated with vaccination and COVID-19. The lack of participation in masking and social distancing can also be somewhat attributed to this sentiment, which is in turn fueled by the so-called “infodemic”, or the misinformation and disinformation about the pandemic circulating in the news and media [27,28].

Interestingly, while women and health science majors were the most hesitant prior to roll-out [29], these subpopulations were among those with the highest vaccine uptake once the vaccine became available [30]. College degree attainment is also observed as one of the significant protective factors against vaccine hesitancy [31], but by far, one of the most compelling factors observed in those opting against getting the COVID-19 vaccine has been political affiliation [21,29].

Students identifying as Republican were 2.5 times less likely to obtain a COVID-19 vaccine [29]. Sun and Monnat found that for every one standard deviation increase in vote share for former President Trump, there is a 6.25% decline in the vaccination rate [32]. Conservative political affiliation was also associated with exposure to negative vaccine perceptions and misinformation on social media, which ultimately damaged the overall vaccination rate [21]. Other significant risk factors include a generally low-risk perception of COVID-19, which is predicted by fewer health-promoting behaviors such as mask-wearing, handwashing, and social distancing [29]. Political ideologies and religiosity were also studied as factors influencing anti-vaccination attitudes [33].

### 1.4. Barriers to Campus Services

Repeatedly, it has been found that the severity of mental illness symptoms is not correlated with mental health service use [34]. A great deal of stigma, as well as prohibitive cost, could be factors in not seeking mental health services [34], as only 21% of students reporting clinical levels of depression and anxiety sought professional help [35]. However, students’ use of counseling services seems to have varied based on several demographic characteristics. Younger students, males, and underclassmen were more likely to use on-campus services, whereas older students, females, and upperclassmen were more likely to use off-campus services [34]. Black, Hispanic, and female students were more likely to seek mental health services overall [34].

Universities have also made efforts to provide technology to students. Across the country, students were loaned laptops and hotspots, and colleges negotiated with vendors to help fill this need [12], but simple access to technology is not enough to bridge the rural digital divide. Hotspots connect to the same cellular networks as cellphones do, and in truly remote areas, being issued just a laptop may not be enough to complete schoolwork [36].

### 1.5. Study Aims

The purpose of our research was to uncover the disparities that arise within college students during large-scale disasters, such as the COVID-19 pandemic. College students’ experiences and attitudes diverge significantly from the general population [37,38], but their associated risks and behaviors are understudied during a public health crisis such as the pandemic.

Most notably, we were interested in the disparities in access to basic necessities as well as academic success. Furthermore, we also wanted to evaluate their risk perception of COVID-19 and their vaccination attitudes. By examining the difficulties this vulnerable population experienced, we plan to better characterize the existing service and academic gaps, hopefully giving rise to more proactive interventions and policy changes at higher educational institutions to prepare for future emergencies and times of instability.

We used a cross-sectional survey design [39] to address and describe participants in terms of the following research objectives:Evaluate the barriers students faced concerning food, housing, and other basic needs during the COVID-19 pandemic.Identify the barriers students faced to academic success in the learning environment during the COVID-19 pandemic.Identify the barriers to and attitudes toward COVID-19 vaccination.Assess the prevalence of COVID-19 among the participants.

## 2. Materials and Methods

### 2.1. Participants and Sampling

Participants were selected using a convenience sample of undergraduate and graduate students attending Tarleton State University [39]. A total of 948 students initially participated in the survey, though 120 responses were removed due to missing data or the participant being under 18. A final sample of 828 students was obtained. Participant demographic characteristics in terms of COVID-19 vaccination status are shown in Table 1. Participants primarily fell within the 18–45-year-old age range and were predominantly female (73%) and White (64%). Participants were mixed in terms of other demographic characteristics such as first-generation student status, school year classification, parental education level, and college.

### 2.2. Data Collection Tool

The data collection tool was an online survey built using the Qualtrics survey platform. The survey included 83 items encompassing a variety of topics such as student demographics, COVID-19 characteristics (e.g., have students contracted it and/or been vaccinated), learning difficulties during the pandemic, social determinants (e.g., food insecurity, job loss), and mental and behavioral health. The behavioral health data were collected using reliable and validated instruments [40,41,42,43]. The present paper focuses on the survey portions pertaining to students’ basic health information, such as prior seropositivity status for COVID-19, vaccination status for COVID-19, as well as academic and basic needs. Although some portions of the survey assessed students’ mental health outcomes (e.g., items from the Patient Health Questionnaire (PHQ-9)) and behavioral health outcomes (e.g., substance use issues), those items are not the focus of the current paper and will be discussed elsewhere. All items examined in this study were developed by two lead research team members for the purpose of this research.

Items examined in the present study covered various topics and used a range of potential response options/formats. One question pertaining to students’ learning difficulties during the pandemic asked “Did you face learning difficulties due to the following learning environments?” and featured the following response options: [a] Online learning, [b] Hyflex (hybrid) learning, [c] Face-to-face learning, [d] Not sure, and [e] Other (please specify). Other survey items discussed topics such as basic needs losses/barriers experienced during the pandemic (e.g., “Did you experience the following during the COVID-19 pandemic?” with response options [a] Loss of access to food, [b] Loss of housing, [c] Loss of job, [d] Other, and [e] None of these), vaccination uptake (e.g., “Have you taken the COVID-19 vaccine?”) and use of services to overcome academic difficulties (e.g., “Did you utilize one or more of the following services to overcome academic difficulties? Select all that apply,” with response options [a] Writing Center, [b] Tutoring Center, [c] Library, [d] Counseling Center, and [e] Other [please specify]). In addition, participants were also asked about their history of COVID-19, as well as where they might have contracted the virus and how they were diagnosed.

### 2.3. Procedures

Between 25 June and 16 July 2021, prospective participants responded to an email sent to their university-affiliated email addresses introducing the study and inviting them to participate. The email contained an anonymous link to the survey, which was developed using Qualtrics, an online survey software. The first page of the survey contained the informed consent form, which described the nature of the study, its procedures, and potential risks and benefits. Participants provided their informed consent by clicking the “continue to the survey” button and proceeding to the next screen, where the survey began. Participants then completed the survey, providing their responses regarding their experiences during the COVID-19 pandemic relevant to their various academic and health outcomes. Some students enrolled in the university’s general psychology course were offered course credit for participating in the study.

### 2.4. Data Management and Analysis

To address the research objectives outlined previously, we analyzed data using SPSS version 27. Because the current study aimed to describe participants in terms of the research objectives, we conducted basic descriptive statistics and/or frequency analyses, as well as some correlational explorations. Although missing data were present throughout the survey responses, they were handled using pairwise deletion to retain as much available data as possible. Furthermore, although missing data can adversely affect the accuracy of inferences made in inferential statistical tests [44], we had a large sample size (*n* = 828) that may have circumvented this issue. The deidentified data were imported from Qualtrics into SPSS for cleaning and final analysis.

## 3. Results

### 3.1. Barriers to Basic Needs

We asked participants about the barriers they faced regarding food, housing, and finances during the pandemic. Specifically, students were asked about their loss of access to a range of needs during the pandemic; frequency results are shown in Figure 1. Overall (*n* = 828), the most commonly experienced resource loss was job loss (26% of participants, or 62.9% of those that experienced resource loss overall). Although less common, several participants also experienced a loss of access to food and housing.

### 3.2. Barriers in the Learning Environment

Several survey items pertained to the barriers students experienced in the learning environment. A descriptive analysis of these items revealed several trends in students’ experiences. Figure 2 demonstrates that online learners had the most difficulty (*n* = 334), Hyflex learners had the second most difficulty (*n* = 111), and face-to-face learners had the least difficulty (*n* = 41). In addition, 42.8% (*n* = 308) of students reported difficulty communicating with the instructors, while 57.2% (*n* = 411) did not.

To examine potential differences across different student groups in terms of their learning environment difficulties, Table 2 shows the results of a series of chi-square analyses comparing the frequencies of learning platform difficulties across various student characteristics. In general, results indicated that a student’s college was the only variable that was statistically significantly related to difficulties in learning environments (*p* = 0.028). An examination of the observed and expected frequencies revealed that more students from the College of Agriculture and Environmental Sciences experienced learning difficulties in the Hyflex environment than expected, and more students from the College of Health Sciences and Human Services and the College of Science and Technology experienced learning difficulties in the online environment than expected. All other chi-square analyses were not statistically significant.

Table 3 presents the frequencies of different kinds of academic challenges students faced during the pandemic. The most common difficulties students encountered included staying motivated (55.7%), working with peers on projects (42.1%), finding a quiet place to learn (33.7%), having high-speed internet access (30.3%), and being too emotionally disturbed to focus (28.9%). Out of the 719 students who responded to questions regarding academic challenges, 37% (*n* = 308) had difficulty communicating with their instructors.

### 3.3. Barriers to Vaccination

Overall, 339 participants (40.9%) reported having received the COVID-19 vaccine, and 326 participants (39.4%) reported having not received it. Participants were also asked about their awareness of the availability of the COVID-19 vaccine from various sources (i.e., on campus, off campus, etc.), irrespective of whether they were vaccinated or not. Of the respondents, 370 participants (44.7%) reported knowing where to get the vaccine on campus, and 535 participants (64.6%) reported knowing where to get the vaccine off campus.

Additionally, unvaccinated participants were asked about their intentions regarding the COVID-19 vaccine, and the frequency results are shown in Figure 3. Overall, among unvaccinated participants, 53.7% of respondents did not plan to get the vaccine, 34.1% were uncertain (i.e., either “Maybe” or “I do not know”), and only 12.8% of respondents planned to get the vaccine in the future. 

Students who indicated “no” were also asked about their reasons for not getting vaccinated, such as not thinking COVID-19 was a threat or lack of knowledge surrounding the virus. Participants’ responses are depicted in Table 4. Overall, the most commonly cited reasons for not getting the vaccine included not thinking COVID-19 was a serious threat (32.2%), lack of knowledge about the health risks of COVID-19 (26.1%), and lack of time to get a COVID-19 vaccine (8.0%).

Of the participants that selected “other reasons” and elected to add a written response, participants’ number one concern (*n* = 30) was about the “safety of the vaccine, [its lack of] FDA approval status, and lack of research” followed by the “hastiness in vaccine development and newness of the vaccine”, (*n* = 24) and their “fear of side effects and complications” (*n* = 10).

A history of COVID-19 (Yes/No) was not associated with the vaccination status (*p* > 0.05) of the study participants (see Figure A1). Of those that were vaccinated and tested positive for COVID-19 (*n* = 84), 54 (64.3%) of them were diagnosed with COVID-19 when they were living with family at home, 16 (19.0%) while living off campus but not at home, and 9 (10.7%) while living in a residence hall (Figure A2 and Figure A3). Participant’s knowledge of locations where they could obtain the vaccine (both on and off campus) is displayed in Figure A4.

### 3.4. Barriers to Campus Resource Use

The final area examined was participants’ knowledge and use of various on- and off-campus services to help meet their academic and basic needs. First, participants indicated the extent to which they used various university services to help meet their academic needs during the pandemic, represented in Table A1. The university library was the most frequently used resource/service at 31.3% of the total sample. Other university services were used at a significantly lower rate.

In addition to the services relevant to students’ academic needs, the survey also included questions regarding the services students used to overcome insecurities relevant to their basic needs. Table A2 depicts these results. Overall, although some resources saw usage among a relatively small number of participants, most participants (65.5% of the total sample) indicated they did not take advantage of any kind of services to help meet their basic needs during the pandemic.

## 4. Discussion

College is often regarded as a transitional period in a young adult’s development. As a time of emerging identity and independence, it is a stepping stone into full adulthood. While the campus environment is not a perfect representation of the real world, it is an intermediate space designed to protect students as they obtain academic and non-academic skills away from the safety of their family homes. Dorms, dining halls, libraries, computer labs, and on-campus work stand in for the real-world challenges of secure housing, home-cooked meals, internet access, and stable employment.

These protections quickly fell through as COVID-19 spread, re-exposing students to the difficulties they were once shielded from. Given students’ vulnerability socially, economically, and academically, it is critical to peel back the layers of interaction between academic settings and the struggles these students face, especially during times of crisis. Therefore, the aim of our study was to uncover the barriers and complexities college students faced during the COVID-19 pandemic. We highlight some determinants that could be potential areas of intervention to improve student success in future disasters.

### 4.1. Barriers to Basic Needs

Social determinants are significant predictors of student success. In the present study, over one in four respondents reported losing work due to the pandemic, representing the most common financial difficulty encountered. Less common financial difficulties reported were food insecurity (8.5%) and housing insecurity (3.6%), and compared to other studies documenting college students’ financial shocks during this time, our participants seemed to experience less difficulty with financial stability [4,7,45,46]. Notably, the majority of the literature findings [4,19,20,21,47], as well as ours, had an overrepresentation of White (63.6%) and female (72.5%) students, but this is consistent with the demographics of the university where the participants were enrolled [48], and across America [49].

### 4.2. Barriers in the Learning Environment

We identified significant barriers to our study population’s ability to adapt to online learning. Consistent with the literature, students faced increasing levels of difficulty as the proportion of learning that took place online increased [20]. Over one-third of participants reported challenges in communicating with their professors, and 40.2% of students experienced difficulties with online learning in general. Fewer students had difficulty learning in the Hyflex model (13.3%), and fewer still found face-to-face learning challenging (5.0%). The primary barrier throughout this period was motivation (55.7%), though other notable challenges included trouble working with peers (42.1%), finding a quiet place to learn (33.7%), and a lack of high-speed internet access (30.3%).

The risk of psychological stress also seems to be contingent on the learning model, with students only taking online classes reporting a higher level of distress compared to their hybrid and face-to-face peers [20]. Similarly, surveys examining students’ ability to cope with and succeed academically following the abrupt shift to online learning found that a third of students found the shift challenging and particularly detrimental to their academic success [19]. These effects were produced not only by so-called “Zoom fatigue” (social exhaustion related to long collaborations and classes over Zoom) but also by a lingering drain on their motivation that reduced their engagement in class [16]. This effect was amplified in students who were in strictly in online learning modalities [19].

Other difficulties such as a disruptive home environment (e.g., difficulty finding a quiet place to learn) (33.7%), balancing home responsibilities and schoolwork (27.7%), having to take care of relatives (15.1%) and siblings (5.7%), general emotional disturbance (28.9%), and lack of internet access (30.3%) are well described in the literature, and our results seem to corroborate these findings [4,7,9,12,50].

### 4.3. Vaccine Hesitancy

Presently, there is ample evidence of students’ risk perception being influenced by their age, sex, race, insurance status, and political affiliation [21,29,51]. Among our unvaccinated participants, 32.2% responded that they did not think that COVID-19 was a threat to their health, 26.1% responded that they did not have enough information about the health risks associated with COVID-19, and 33.4% responded “other reasons”. Free responses from participants detailed concerns regarding its safety and newness and contentious opinions regarding the government. These are commonly cited reasons for COVID-19 vaccine hesitancy [52,53], but other reasons found in the literature included low-risk perception and exposure to misinformation [54]. Furthermore, belief in conspiracy theories (e.g., referencing a lack of belief in the pandemic or a distrust of the government), concern about governmental overreach, or existing anti-vaccine beliefs also remain attitudes concerning to public health [29,55].

On the other hand, a few studies have indicated that students, while often not as concerned about their health, were significantly more concerned with the health of their friends and family [56,57]. It is possible that our participants believed that COVID-19 was only a threat to those who are immunocompromised or of advanced age [58]. They may also have received messaging from spaces that are wary of vaccines or the scientific community in general. Since students who lean conservative are more vulnerable to this messaging, and anti-vaccine social media is much more amplified and readily found than factual public health information [21,55], this could also explain part of this response. Similarly, those responding that they did not have enough information about the health risks associated with COVID-19 could have felt that their information did not substantiate a high enough risk to get vaccinated. This lack of information could also be influenced by social media and low access to healthcare [21,32,51,55,57].

We discovered that even though nearly 1 in 5 participants reported testing positive for COVID-19 in the past, only 53.2% of those participants had gotten the vaccine despite there being availability on and off campus at no cost. A total of 32.2% of unvaccinated participants indicated that they did not believe COVID-19 was a threat to their health, and 26.1% reported that they did not know about the health risks associated with the virus. Further, about 10% thought it would cost them to get the vaccine. The US Department of Health and Human Services (HHS) has been consistent in its messaging that COVID-19 vaccines are safe, free, and effective [59], and as of writing, the COVID-19 vaccine is still free for Americans even after the end of the public health emergency [60,61]. Many credible agencies, such as HHS, publish resources and information that students may not be aware of.

Together, this not only suggests that many students underestimated the risk of long-term COVID-19-related morbidity and mortality for themselves, but also underscores that vaccine education must be prioritized on college campuses, highlighting the risks versus benefits before and after the release of novel vaccines during large outbreaks.

Predictors of vaccine compliance include parental educational attainment, insurance status, and college degree attainment, among other factors [22,31]. In our study, only age, student classification, and health insurance were associated with vaccine uptake. Vaccination status was not associated with a history of COVID-19 positivity (Figure A1). Participants’ residential status at the time of diagnosis and location of probable or suspected exposure seemed to follow the same trend regardless of vaccination status (Figure A2 and Figure A3).

The literature also indicates many infectious outbreaks are associated with social stigma. This was true for COVID-19 as well [62,63]. Approximately 22.7% of our study population reported experiencing social stigma either at college or in their community due to their COVID-19 positivity status.

### 4.4. Barriers to Campus Services

Over 85% of participants indicated that they knew where to get academic assistance, but relatively few accessed these services. About one in three (31.3%) participants reported using the campus library, but other services were used substantially less (Tutoring and Learning Center (12.3%), Student Counseling Services (10.1%), and Writing Center (9.3%)). Since there is little examination of campus resource use during the pandemic, comparing our results to the college student population at large is difficult. However, it does seem that our participants used these resources less than students described in the literature. Lee et al. [34] reported twice the rate of counseling services usage in their study, but there might have been differences in the sample demographics. White, female, and older students are overrepresented in our sample, and Lee et al. seemed to have had a more heterogeneous sample.

Just over a third (35.5%) used campus services to meet their basic needs. However, two thirds of the participants lived off campus during the study period, which could explain the underutilization of the many on-campus resources, such as the food pantry. Even so, despite 26.1% of participants reporting job loss, only 4.3% overall utilized the career services, even when their services could be accessed remotely. This warrants further investigation, as it implies students did not see career services as a resource to find employment and stabilize their financial situation. It could very well be that students did not feel compelled to find a job because they were financially secure enough at home. Alternatively, it may indicate that career services could not sufficiently meet the needs of students who had lost their jobs or that students may not have been aware that career services were still open during the pandemic for help with employment opportunities. Since those working part-time or informal jobs were hit the hardest by job loss [7,64], this might demonstrate an area of growth for university career development services in general.

## 5. Limitations and Strengths

### 5.1. Limitations

Our research uniquely focuses on college students attending a public university whose main campus is in central Texas and whose satellite campuses are in north Texas. Although nearly every county in the state is represented in the university [65], the findings of our study may not be generalizable to public universities in different states, as well as historically Black colleges or universities or Hispanic Service Institutions. Our sample’s characteristics, like many others describing college students during this time, had an overrepresentation of White and female students, which is also consistent with the demographic breakdown of our university [48,66].

While we attempted to characterize the unique barriers students faced with respect to campus resources and general health during the pandemic, several inherent methodological limitations remain. The cross-sectional design and self-selection by study participants may have compromised the generalizability of our findings. While self-reported data are more likely to yield results that are impacted by social desirability bias, anonymity likely provided more truthful answers. However, it should be noted that due to how the data were coded concerning analyses involving binary variables where 1 = participant responded to an item, and 0 = participant did not respond to an item, “0” responses contain both participants that did not respond deliberately and missing data.

Political affiliation has been studied as a potential factor for vaccine hesitancy [23,30,67], but we did not investigate this possible factor to illuminate its effect on COVID-19 vaccination status. Further, our cross-sectional design may have created a bias toward underreporting, given that those who may not have the time or technology to respond may not have responded [68]. This might have explained the comparatively low food and housing insecurity. Other factors influencing participation could have been strong beliefs about COVID-19 (either negative or positive) and receiving course credit. Lastly, with regard to the proportions pertaining to first-generation students (Table 1), there are inconsistent approaches to how this variable is defined in the literature [69]. Readers should exercise caution when interpreting the proportions.

### 5.2. Strengths

Because all our data were collected at once, there was no probability of a loss to follow-up, and there is a vast amount of data that could be analyzed to further explore multiple exposures and outcomes in the midst of the pandemic. Using convenience sampling within our cross-sectional design allowed us to quickly evaluate the prevalence of certain characteristics of students attending a mid-sized public university to inform decisions at multiple levels, from instruction to administration. Cross-sectional designs have tremendous value for exploratory analysis and in ruling out many probable associations [70]. Further, our study provides an excellent platform for generating hypotheses to conduct more robust longitudinal studies that inform policies and improve university responses in the future. By understanding where the response and resources may have been lacking during the pandemic, the university will be better able to identify key areas of improvement to protect college students’ resilience during times of crisis and improve vaccine education efforts before and during the pandemic.

## 6. Conclusions

Our study provides valuable insights concerning college students’ barriers to access during the COVID-19 pandemic and satisfaction with online learning, financial and food insecurities, risk perception for COVID-19, and their associated vaccination status. The underutilization of specific resources, such as career services, underscores a need to improve communication regarding campus services and provide help during a time of crisis. Further, our study highlights the need for educational interventions to emphasize the importance of infectious diseases as well as routine and newly released vaccines for outbreak-related illnesses to increase overall vaccine compliance among college students. Researchers interested in the subject area should design robust longitudinal studies to understand the long-term effects of COVID-19 related to basic needs, insecurities, infection risk perception, vaccine compliance, and learning difficulties. This understanding will help develop and implement future disaster-specific interventions during public health emergencies.

## Figures and Tables

**Figure 1 ijerph-20-06924-f001:**
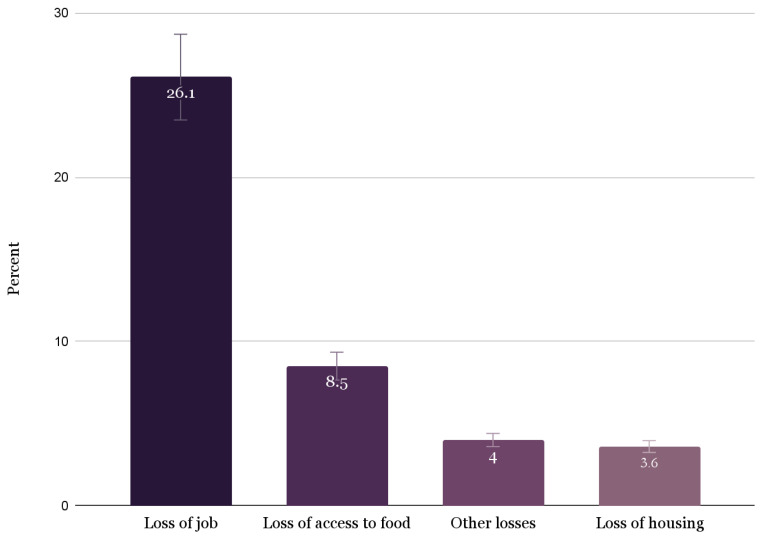
Challenges experienced by students regarding basic needs (*n* = 343) [Note: students could report to more than one category].

**Figure 2 ijerph-20-06924-f002:**
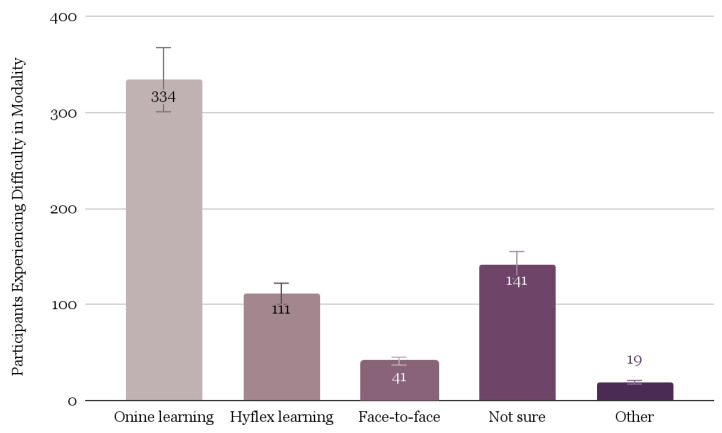
Comparison of difficulty students experienced by learning modality (Note: students could report to more than one category).

**Figure 3 ijerph-20-06924-f003:**
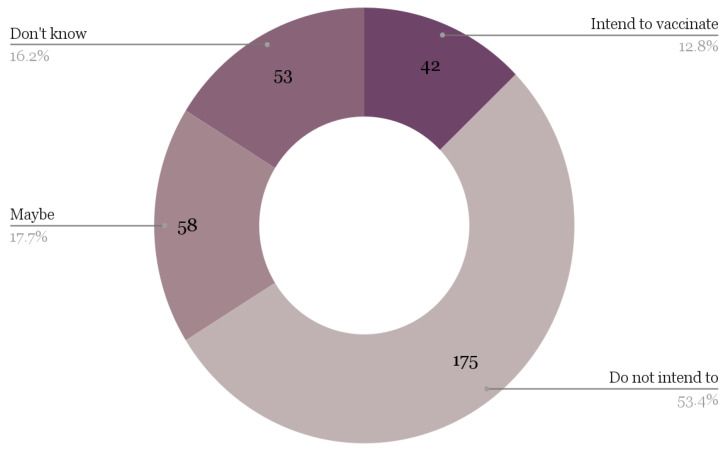
Unvaccinated participants’ intentions to vaccinate (*n* = 370).

**Table 1 ijerph-20-06924-t001:** Participants’ Selected Demographic Characteristics and COVID-19 Vaccination Status.

Variable	Sample (*n* = 828)		Vaccinated (*n* = 339, 40.9%)			Unvaccinated (*n* = 326, 39.4%)			Chi-Square *p*-Value
	*n*	(%)	*n*	Vaccinated (%)	Sample (%)	*n*	Unvaccinated (%)	Sample (%)	
Age									0.004 *
18–45 years	724	87.4	287	84.7	34.7	298	91.4	36.0	
>45 years	36	4.3	24	7.1	2.9	8	2.5	1.0	
Missing	68	8.2	28	8.3	3.4	20	6.1	2.4	
Sex									0.400
Male	206	24.9	85	25.1	10.3	82	25.2	9.9	
Female	600	72.5	242	71.4	29.2	238	73.0	28.7	
Other	14	1.7	8	2.4	1.0	3	0.9	0.4	
Do not wish to disclose	8	1.0	4	1.2	0.5	3	0.9	0.4	
Ethnicity									0.018 *
Caucasian	527	63.6	196	57.8	23.7	215	66.0	26.0	
African American	98	11.8	48	14.2	5.8	36	11.0	4.3	
Asian	11	1.3	5	1.5	0.6	4	1.2	0.5	
Hispanic	148	17.9	78	23.0	9.4	48	14.7	5.8	
Other	43	5.2	12	3.5	1.4	23	7.1	2.8	
Missing	1	0.1	0	0.0	0.0	0	0.0	0.0	
^#^ First Generation Student									0.665
Yes	384	46.4	156	46.0	18.8	145	44.5	17.5	
No	442	53.4	182	53.7	22.0	181	55.5	21.9	
Missing	2	0.2	1	0.3	0.1	0	0.0	0.0	
Classification									0.001 *
Freshman	102	12.3	22	6.5	2.7	26	8.0	3.1	
Sophomore	124	15.0	43	12.7	5.2	58	17.8	7.0	
Junior	186	22.5	60	17.7	7.2	92	28.2	11.1	
Senior	203	24.5	107	31.6	12.9	69	21.2	8.3	
Graduate	194	23.4	99	29.2	12.0	70	21.5	8.5	
Postgraduate	15	1.8	6	1.8	0.7	9	2.8	1.1	
Missing	4	0.5	2	0.6	0.2	2	0.3	0.2	
^ Parental Education									0.109
Some high school	94	11.4	50	14.7	6.0	29	8.9	3.5	
High school	141	17.0	53	15.6	6.4	61	18.7	7.4	
Some college	171	20.7	62	18.3	7.5	70	21.5	8.5	
Associate’s degree	68	8.2	28	8.3	3.4	29	8.9	3.5	
Bachelor’s degree	223	26.9	86	25.4	10.4	97	29.8	11.7	
Master’s degree	100	12.1	48	14.2	5.8	32	9.8	3.9	
Doctorate degree	17	2.1	6	1.8	0.7	6	1.8	0.7	
Missing	14	1.7	6	1.8	0.7	2	0.6	0.2	
College									0.091
Agricultural and Environmental Sciences	137	16.5	45	13.3	5.4	69	21.2	8.3	
Business	139	16.8	62	18.3	7.5	59	18.1	7.1	
Education	139	16.8	65	19.2	7.9	45	13.8	5.4	
Health Science and Human Services	182	22.0	71	20.9	8.6	69	21.2	8.3	
Liberal and Fine Arts	118	14.3	49	14.5	5.9	41	12.6	5.0	
Science and Technology	112	13.5	47	13.9	5.7	42	12.9	5.1	
Missing	1	0.1	0	0.0	0.0	1	0.3	0.1	
Health Insurance									0.021 *
Insured	661	79.8	283	83.5	34.2	248	76.1	30.0	
Uninsured	164	19.8	56	16.2	6.7	77	23.6	9.3	
Missing	3	0.4	0	0.0	0.0	1	0.0	0.0	
Residence									0.124
On-campus	274	33.1	95	28.0	11.5	109	33.4	13.2	
Off-campus	552	66.7	244	72.0	29.5	216	66.3	26.1	
Missing	2	0.2	0	0.0	0.0	1	0.0	0.0	

* Statistically significant at *p* < 0.05; ^#^ biological parents do not have a college degree; ^ at least one parent’s highest educational level.

**Table 2 ijerph-20-06924-t002:** Chi-square results comparing learning environment difficulties across participant characteristic variables.

Variable	Online		Hyflex		Face-to-Face		Not Sure		Other		Chi-Square *p*-Value
	*n*	(%)	*n*	(%)	*n*	(%)	*n*	(%)	*n*	(%)	
Age											0.418
18–45 years	296	49.0	101	16.7	35	5.8	128	21.2	44	7.3	
>45 years	11	47.8	2	8.7	1	4.3	5	21.7	4	17.4	
Sex											0.277
Male	93	53.4	26	14.9	5	2.9	36	20.7	14	8.0	
Female	231	47.6	79	16.3	34	7.0	104	21.4	37	7.6	
Other	6	54.5	3	27.3	0	0.0	1	9.1	1	9.1	
Do not wish to disclose	3	37.5	2	25.0	2	25.0	0	0.0	1	12.5	
Ethnicity											0.133
Caucasian	199	47.2	72	17.1	19	4.5	96	22.7	36	8.5	
African American	38	45.8	10	12.0	10	12.0	20	24.1	5	6.0	
Hispanic	71	53.0	24	17.9	10	7.5	21	15.7	8	6.0	
Other	25	64.1	4	10.3	2	5.1	4	10.3	4	10.3	
^#^ First Generation Student											0.080
Yes	148	47.7	45	14.5	17	5.5	74	23.9	26	8.4	
No	184	50.1	65	17.7	24	6.5	67	18.3	27	7.4	
Classification											0.059
Freshman	45	60.8	5	6.8	2	2.7	19	25.7	3	4.1	
Sophomore	52	45.2	24	20.9	8	7.0	26	22.6	5	4.3	
Junior	78	45.1	30	17.3	14	8.1	39	22.5	12	6.9	
Senior	100	52.9	32	16.9	12	6.3	27	14.3	18	9.5	
^ Parental Education											0.867
Some high school	41	51.2	10	12.5	6	7.5	17	21.3	6	7.5	
High school	55	46.6	20	16.9	5	4.2	26	22.0	12	10.2	
Some college	61	45.5	19	14.2	10	7.5	36	26.9	8	6.0	
Associate’s degree	26	49.1	9	15.8	5	8.8	10	17.5	5	8.8	
Bachelor’s degree	96	52.5	34	18.6	10	5.5	31	16.9	12	6.6	
Master’s degree	38	46.3	17	20.7	4	4.9	18	22.0	5	6.1	
Doctorate degree	6	42.9	1	7.1	1	7.1	3	21.4	3	21.4	
College											0.028 *
Agricultural and Environmental Sciences	51	43.2	29	24.6	9	7.6	26	22.0	3	2.5	
Business	53	43.8	15	12.4	9	7.4	32	26.4	12	9.9	
Education	46	43.0	19	17.8	5	4.7	29	27.1	8	7.5	
Health Science and Human Services	80	55.9	18	12.6	10	7.0	23	16.1	12	8.4	
Liberal and Fine Arts	42	46.2	16	17.6	4	4.4	18	19.8	11	12.1	
Science and Technology	61	62.9	13	13.4	4	4.1	13	13.4	6	6.2	
Health Insurance											0.124
Insured	261	48.1	87	16.0	29	5.3	120	22.1	46	8.5	
Uninsured	72	53.7	23	17.2	12	9.0	20	14.9	7	5.2	
Residence											0.091
On-campus	113	51.6	39	17.8	12	5.5	45	20.5	10	4.6	
Off-campus	218	47.7	71	15.5	29	6.3	96	21.0	43	9.4	

* Statistically significant at *p* < 0.05; ^#^ biological parents do not have a college degree; ^ at least one parent’s highest education level.

**Table 3 ijerph-20-06924-t003:** Frequency Information for the Types of Difficulties Students Experienced.

Reason for Not Getting Vaccine	*n*	%
Staying motivated to learn	461	55.7
Interacting with peers for group projects and assignments	349	42.1
Finding a quiet place to learn	279	33.7
Having high-speed internet access/connection	251	30.3
Being too emotionally disturbed to focus on academics	239	28.9
Having difficulty balancing household responsibilities with academics	229	27.7
Having difficulty balancing work with academics	217	26.2
Having a good quality microphone or camera on my computer	131	15.8
Not knowing where to get help for academic success	131	15.8
Having to take care of relatives or family members during the pandemic (but not related to COVID-19)	125	15.1
Inability to find student support services when needed	99	12.0
Being too physically unwell to focus on academics (but not related to COVID-19)	94	11.4
None of these options interfered with my learning	90	10.9
Having been diagnosed with COVID-19 myself	79	9.5
Having to take care of relative or family member that had COVID-19	69	8.3
Having to babysit my siblings	47	5.7
Other	26	3.1

Note: Percentage values represent the proportion of the full sample.

**Table 4 ijerph-20-06924-t004:** Frequencies for Unvaccinated Participants’ Reasoning for Not Getting Vaccinated.

Reason for Not Getting Vaccine	*n*	%
Other reasons (concerns about safety and speed of vaccine trials)	109	33.4
I do not think COVID-19 is a serious threat to my health	105	32.2
I do not have enough knowledge about the health risks associated with COVID-19	85	26.1
I am aware of the health risks associated with COVID-19, but the cost of vaccination is too high	31	9.5
I have a busy schedule and do not have time to get the COVID-19 vaccination	26	8.0
I do not know where to get the vaccination on campus	7	2.1
I do not know where to get the vaccination off campus	7	2.1
My insurance does not cover the COVID-19 vaccination	5	1.5

Note: Percentage values represent the proportion of the full sample.

## Data Availability

Data will be available upon request from the corresponding author.

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
