# Peer review of "Identifying Barriers to Basic Needs, Academic Success, and the Vaccination Pattern among College Students during the COVID-19 Pandemic"

_ijerph, 2023, doi:10.3390/ijerph20206924_

Round 1
Reviewer 1 Report
The issue of the manuscript is very interesting (rurally placed universities). It is worth revealing the analysed fields (vaccination status, academic barriers, financial barriers etc.) in these institutions during the pandemic years because this student body has got specific features. The number of the respondents is high enough which can be run more complicated statistics. I think the bases of a good quality paper are given but the authors have to make important improvements.
- the location of the analysis is only one institution. Due to this fact this is rather a field research and we have to clearly see the situation and the features of the given institution. To sum up, the description of the location has to be replaced.
- If the issue of the analysis is the rurally placed university, it is worth creating a separated chapter for this phenomena. The ‘rural student’ is another category because there are rural students at the elite universities too.
- the line spacing is not suitable in the row 207-216
- gender or sex? If I know right it is important to clarify these notions at this journal.
- first generation students – authors have to clarify this too. If the parents’ educational level were different (one has got a degree and the other has not) what was the category of the respondents? (Table 1).
- Parental educational level – same remark. Does this value belong to the father or mother?
- Are there any information about the type of the settlement? I suppose that there are differences in the case of this variable.
- I can see statistical test in the case of Table 1, but the authors should to use tests in later phase too – the number of the respondents is high enough. E.g. factors and clusters can be identified according to data (Table 4). The authors can use variance analysis or cross-tabs - there are a lot of possibilities.
- The analysed fields can be compared according to the different subsamples (sex, first generation students etc.).
- It is easier to interpret the data if the items in the tables are arranged in descending or ascending order.
- In the case of the enlargement of the theoretical frames the following phenomena can be used: anti-intellectualism (in the case of the attitude toward vaccination) and there is a relatively wider theoretical background for risk perception.
- I think something is wrong in the row 354.
- Research objectives can be more elaborated.
To sum up, in my opinion major changes are needed but the bases are given. In my opinion these statistics are a little bit simple for this journal.
Author Response
Thank you for giving us the chance to revise our paper. Please find attached our detailed responses to your suggestions.
Sincerely,
S. Gandhi

Reviewer 2 Report
It is important to investigate the effect of the pandemic on rural college students.
The sample size is appropriate for the research questions.
Some mention of the limitations of using respondents from a single institution should be made. Texas is a place with a particular distribution of politics and also a place where people can be outside in ventilated areas more of the year than other regions of the country. How representative are Texas institutions of rural institutions country wide? Would the vaccination rate or unemployment rate due to Covid have looked different in other regions of the country?
This study needs improvement in terms of what it offers as implications. Are the authors expecting another pandemic? Is the study about how to prepare for the next pandemic? Is the study about how to deal with the ongoing risk of Covid? Is the paper about how to help students who have decreased funding and diminished learning because of the Covid pandemic? What are the implications for universities? What are the implications for policy making at the state or federal level?
Author Response

(The authors gave the same response as above.)

Reviewer 3 Report
Many thanks for your manuscript. Below are my comments and suggestions to help you make it better.
(1) Abstract: More information about the research methods should be presented, specifically, the sample size and sampling techniques.
(2) Keywords: COVID-19, college students and vaccination are not good keywords as they repeat exactly words/phrases of the paper title.
(3) Introduction: This current section includes the Literature review section, which should be separated. Moreover, the research questions should be presented in the Introduction.
(4) Materials and Methods:
- References used to develop the survey questionnaire should be described.
- Did you pilot the instrument?
- How did you ensure the reliability and validity of the instrument?
(5) Results: Only descriptive analysis was presented. I think you need inferential analysis as well.
It should be proofread.
Author Response

(The authors gave the same response as above.)

Round 2
Reviewer 1 Report
Dear Authors,
thank you for your detailed answer.
Clarification:
Are there any information about the type of the settlement?
This question referred to the attributes of the type of settlement (capital city, village, suburban area etc.). I think this independent variable can form the attitude of students – in a rural college too. This sentence was not clear-cut so it was my fault.
Author Response
Thank you so much for your feedback. Please see the attachment.
S Gandhi

Reviewer 2 Report
The discussion about the generalizability of a sample from a Texas institution has improved the paper.
The discussion section continues to need improvement. It currently reads like a report of results rather than an exploration of the implications of the findings for educational institutions and policy makers.
Author Response

(The authors gave the same response as above.)

Reviewer 3 Report
Many thanks for your effort to revise the manuscript. I do not have any comment at this moment.
The quality of English language is good.
Author Response
Thank you so much for reviewing our paper. We really appreciate your time.
S Gandhi